# Adulthood Socioeconomic Position and Type 2 Diabetes Mellitus—A Comparison of Education, Occupation, Income, and Material Deprivation: The Maastricht Study

**DOI:** 10.3390/ijerph16081435

**Published:** 2019-04-23

**Authors:** Yuwei Qi, Annemarie Koster, Martin van Boxtel, Sebastian Köhler, Miranda Schram, Nicolaas Schaper, Coen Stehouwer, Hans Bosma

**Affiliations:** 1Department of Social Medicine, Maastricht University, P.O. Box 616, 6200 MD Maastricht, The Netherlands; y.qi@umcg.nl (Y.Q.); a.koster@maastrichtuniversity.nl (A.K.); 2CAPHRI School for Public Health and Primary Care, Maastricht University, 6200 MD Maastricht, The Netherlands; n.schaper@mumc.nl; 3Department of Psychiatry and Neuropsychology, School for Mental Health and Neuroscience, Maastricht University Medical Centre, 6200 MD Maastricht, The Netherlands; m.vanboxtel@maastrichtuniversity.nl; 4School for Mental Health and Neuroscience (MHeNS), Maastricht University, 6229 ER Maastricht, The Netherlands; s.koehler@maastrichtuniversity.nl; 5Department of Medicine, Maastricht University Medical Centre+, 6229 HX Maastricht, The Netherlands; m.schram@mumc.nl (M.S.); cda.stehouwer@mumc.nl (C.S.); 6Cardiovascular Research Institute Maastricht (CARIM), Maastricht University, 6229 ER Maastricht, The Netherlands; 7Department of Internal Medicine, Maastricht University Medical Centre, Randwycksingel 35, 6229 EG Maastricht, The Netherlands

**Keywords:** socioeconomic position, type 2 diabetes mellitus, prediabetes, epidemiology

## Abstract

In an effort to better quantify the impact of adulthood socioeconomic circumstances on prediabetes and type 2 diabetes (T2DM), we set out to examine the relative importance of four adulthood socioeconomic indicators. Using cross-sectional data from The Maastricht Study on 2011 middle-aged older men and women, our findings indicate that low educational level (OR = 1.81, 95% CI = 1.24–2.64), low occupational level (OR = 1.42, 95% CI = 0.98–2.05), and material deprivation (OR = 1.78, 95% CI = 1.33–2.38) were independently associated with T2DM. Low income (OR = 1.28, 95% CI = 0.88–1.87) was the strongest, albeit not significant, SEP (socioeconomic position) correlate of prediabetes. This association confirms SEP as a multifaceted concept and indicates the need to measure SEP accordingly. In order to tackle the social gradient in prediabetes and T2DM, one should, therefore, address multiple SEP indicators and their possible pathways.

## 1. Introduction

Different indicators of adulthood socioeconomic position (SEP) have been used in studies that investigated the socioeconomic gradient in type 2 diabetes mellitus (T2DM). Most have used education or occupation, others used income, material wealth, or ownership of various material assets [1,2,3,4,5,6,7,8,9,10]. The indicators generally show inverse associations with T2DM, however, with different strengths. Stronger direct influences of either indicator in such a research effort might point to the importance of specific pathways toward T2DM. Rather than emphasizing the broad influence of SEP, recent evidence suggests that the direct influence of different indicators of SEP, thus aside from being related with each other during the life course, may influence T2DM through different pathways [11]. For example, education indicates the acquisition of knowledge on health behaviors, occupation relates to prestige and work exposures, while income reflects material living resources [1,10,12,13,14]. By pointing to such pathways, information on the relative importance of different SEP indicators for T2DM might be of help in designing better policies or interventions to address the socioeconomic gradient in T2DM [15,16].

Using cross-sectional data from The Maastricht Study on middle-aged and older men and women, we set out to examine the relative (direct) contribution of four adulthood SEP indicators in relation to prediabetes and T2DM. As socioeconomic disadvantage in early life might also be associated with a higher risk of both T2DM and low adult SEP, the associations of adulthood SEP were studied, independent of early life SEP [17]. Furthermore, as some argue that intellectual abilities are the driving force behind socioeconomic (educational in particular) differences in health, the present study will also take account of differences in verbal intelligence [18,19]. Last but not least, we incorporated prediabetes, which is an intermediate state of hyperglycemia with glycemic parameters above normal but below the T2DM threshold [20]. Knowing the risk factors for prediabetes can help prevent or delay its progression towards T2DM.

## 2. Materials and Methods

### 2.1. Study Population

Data came from The Maastricht Study, an observational prospective population-based cohort study. Detailed rationale and methods have been described previously [21]. In short, The Maastricht Study focuses on the etiology, pathophysiology, complications, and comorbidities of T2DM. The study is characterized by an extensive phenotyping approach. Eligible participants were aged between 40 and 75 years and living in the southern part of The Netherlands. Participants were recruited through mass media campaigns and from the municipal registries and the regional Diabetes Patient Registry via mailings. Recruitment was stratified according to known T2DM status for reasons of efficiency. The current study included cross-sectional data from the first 3451 participants who completed the baseline survey between November 2010 and September 2013. Participants with type 1 diabetes (*n* = 33) and other types of diabetes (*n* = 4) were excluded first. Subsequently, participants who did not report their education information (*n* = 77), job information (*n* = 627), income information (*n* = 879), or early life SEP (*n* = 22) were excluded. A total of 2,011 participants were included in the analyses.

### 2.2. Measures

#### 2.2.1. Diabetes Outcome

T2DM was defined according to the World Health Organization (WHO) diagnostic criteria of glucose tolerance status [22]. All participants underwent a standardized seven-point oral glucose tolerance test (OGTT) after overnight fasting. Blood samples were collected at baseline, and 15, 30, 45, 60, 90 and 120 min after consumption of the 75 g glucose drink. Participants who were insulin-dependent and participants with a fasting glucose level higher than 11.0 mmol/L (as determined by finger prick) did not undergo this test. Prediabetes was defined as IFG (fasting plasma glucose 6.1–6.9 mmol/L and 2-h plasma glucose <7.8 mmol/L), IGT (fasting plasma glucose <7.0 mmol/L and 2-h plasma glucose ≥7.0–<11.1 mmol/L) or both. T2DM was defined by fasting plasma glucose ≥7.0 mmol/L or 2-h plasma glucose ≥11.1 mmol/L in accordance with WHO 2006 criteria. Participants on diabetes medication and without type 1 diabetes were also considered as having T2DM.

#### 2.2.2. Educational Level

The educational level of the participant was assessed by questionnaire with eight categories (i.e., 1. No education, 2. Primary education, 3. Lower vocational education, 4. General secondary education, 5. General vocational education, 6. Higher secondary and pre-university education, 7. Higher vocational education and 8. University). For this study, three categories were created for educational level: low (1–3), middle (4–6), and high (7 and 8).

#### 2.2.3. Occupational Level

Participants were asked to describe their current or previous job. By using the International Standard Classification of Occupations 2008 (ISCO-08), which is a hierarchical classification system based on education and skills required in a job, the job descriptions were classified [23]. The resulting codes were then converted to the International Socio-Economic Index of Occupational Status (ISEI-08) [24,25]. ISEI-08 classifications were categorized as low, intermediate and high occupational status based on tertiles.

#### 2.2.4. Income Level

Income was measured by self-reported net household income per month, consisting of 19 categories, ranging from 0 to >5000 euros per month. To compute the equivalent income level, household size was taken into account by dividing the net household income (midpoints of categories) by the square root of household size [26,27]. By using tertiles, the equivalent income was categorized into low, intermediate and high.

#### 2.2.5. Material Deprivation

Material deprivation was measured by the instrument developed by The Netherlands Institute for Social Research [28]. For the purpose of the present study, four subscales were created: (i) lack of basic goods (range: 0–7) (i.e., lack of one or more of the following items due to financial reasons: freezer, refrigerator, car, oven, washing machine, own house and telephone), (ii) arrears of payment (range: 0–3) (i.e., one or more of the following arrears of payments: mortgage or rent, utility bills and hire purchase instalment), (iii) economic strain (range: 0–6) (i.e., could not afford one or more of the following items or activities: week long holiday away from home, meal with meat, chicken or fish every second day, keep home adequately warm, buy new furniture when needed, buy new clothes when needed, invite family or friends for dinner), and (iv) perceived financial problems (range: 0–3) (i.e., living expenses are considered from not heavy to very heavy and/or managing with household income is considered from very easy to very difficult and/or reimbursement of debts is considered from not heavy to very heavy) [29]. In order to compute an overall measure for material deprivation, all four subscales were recoded into variables with the same range (0–10). The subscales were averaged into a final score. Using tertiles, the final score was recorded into materially deprived, intermediate, and not deprived.

### 2.3. Covariate

Age (years), sex (man, women), early life SEP, and verbal intelligence were covariates. Early life SEP was measured with poverty in youth and educational level of both parents. Poverty in youth was measured with the following question: “Was the financial situation at your childhood’s home sometimes such that there was not enough money to buy food or to replace outworn clothes or shoes?”, with four answering categories: “no, never; yes, sometimes; yes, often; yes, always”. The educational level of the parents consisted of eight categories, ranging from no education to university education (i.e., 1. No education, 2. Primary education, 3. Lower vocational education, 4. General secondary education, 5. General vocational education, 6. Higher secondary and pre-university education, 7. Higher vocational education and 8. University) [17]. To create an overall index of early life SEP, the three variables were standardized and subsequently averaged. By using tertiles, the scores were categorized into low, intermediate and high [17]. Verbal intelligence was measured with the abbreviated version (i.e., vocabulary test section) of the Groningen Intelligence Test (GIT) [30,31].

### 2.4. Statistical Analysis

Spearman correlations were computed to examine the associations between the SEP indicators. Chi2 tests and analysis of variance (ANOVA) were used to examine differences between participants with normal glucose metabolism, prediabetes, and T2DM for the categorical and continuous SEP indicators, respectively. We performed multinomial logistic regression analyses to examine the association between different SEP indicators and diabetes status, using normal glucose metabolism as the reference group. We first estimated these associations in model 1 for each separate SEP indicator, adjusting for sex and age. Then we additionally adjusted for early life SEP in model 2. In model 3, verbal intelligence was additionally adjusted for. Finally, in model 4 the SEP indicators were mutually adjusted for each other. To get additional insight in the relative importance of the SEP indicators, we also did the analyses with the standardized Z scores of each of the continuous SEP indicators. To study their relative direct influence, odds ratios (ORs) and *p*-values were compared between the (standardized and unstandardized) SEP indicators. Sensitivity analyses were done in which the above approach was repeated with linear regression analyses for the hemoglobin A1c (HbA1c) and glucose levels. Furthermore, a compositional SEP indicator was computed by averaging the four standardized SEP indicators and categorizing the result using tertiles. Associations with *p* < 0.05 were considered statistically significant. Although prediabetes and T2DM were part of the same outcome in the multinomial logistic regression, results are presented in separate tables for prediabetes and T2DM. All analyses were conducted using IBM SPSS software version 25.0 (IBM Corp., Armonk, NY, USA).

## 3. Results

The study population consisted of 2011 participants with a mean age of 58.9 ± 8.1 years, of whom slightly less than half were women (48%). As expected, the Spearman correlations indicated that all SEP indicators were related to each other, the strongest correlation being between educational level and the ISEI occupational level (*r* = 0.583). Table 1 presents the descriptive characteristics according to diabetes status. A total of 1232 (61.2%) participants had normal glucose metabolism, 312 (15.5%) had prediabetes, and 467 (23.2%) had T2DM. The descriptive analysis also revealed graded associations between T2DM and all SEP indicators. Participants with T2DM were older, more often men, were lower educated, had lower occupational level, earned less, and more material deprived than participants with normal glucose metabolism or prediabetes.

A lower income was statistically significantly associated with higher odds of prediabetes. When adjusted for sex and age (Table 2, model 1), those with a low-income level had 1.49 higher odds (*p* = 0.01, 95% CI = 1.09–2.04) of prediabetes compared with those with a high income. When adjusted for early life SEP and verbal intelligence (model 3), the association between income level and prediabetes was only slightly reduced to 1.42 (*p* = 0.04, 95% CI = 1.02–1.99). When including all four SEP indicators into the same model (model 4), each SEP indicator lost its statistical significance. Comparing the odds ratios for both the categorical and standardized scores variant and their respective *p*-values in model 4, income level held its strongest associations with prediabetes.

All four SEP indicators were statistically significantly associated with T2DM (Table 3). Participants with a low educational level had a 2.53 (*p* < 0.001, 95% CI = 1.92–3.35) times higher odds of T2DM compared with participants with a high educational level (Table 3, model 1). The addition of early life SEP and verbal intelligence did not result in a significant reduction of the odds (OR = 2.32, *p* < 0.001, 95% CI = 1.66–3.22). We found a similar pattern for occupational level, income level, and material deprivation. For income level and occupational level (but not the standardized variant), the inclusion of all SEP indicators in model 4 attenuated their association with T2DM to non-significance. Participants with a low educational level had a 1.81 (*p* < 0.01, 95% CI = 1.24–2.64) higher odds of T2DM, while materially deprived participants had a 1.78 (*p* < 0.001, 95% CI = 1.33–2.38) higher odds of T2DM compared to their better-off counterparts. The standardized scores of educational level, occupational level, and material deprivation showed the strongest associations with T2DM in the final model.

As for the compositional SEP indicator, participants with the lowest compositional SEP had a strikingly 3.37 higher odds (95% CI: 2.42–4.71) of T2DM compared to their better-off counterparts (adjusted for sex, age, early life SEP, and verbal intelligence) (Figure 1). The odds of having prediabetes for people with the lowest compositional SEP was 1.51 (95% CI: 1.06–2.16). Further in-depth analyses indicated that there were no interaction effects of any of the four SEP indicators with sex, age or early life SEP. No convergence problems were encountered in the full multivariable models. Using the standardized Z scores of each of the continuous SEP indicators, including multiple regression analyses for the HbA1c and glucose level, did not result in a different pattern of findings.

## 4. Discussion

To our knowledge, this is the first study to compare, in a single sample, the relative strengths of the association of various SEP indicators with objectively measured T2DM. Our findings, based on Dutch data, suggest that independent of early life SEP and verbal intelligence, low educational level, low occupational level, and high material deprivation were independently associated with T2DM. The findings were not significant for prediabetes in the fully adjusted model, but low income still had the highest odds ratio (OR = 1.28). The findings suggest that for a full appreciation of the influence of SEP one cannot rely on one indicator only. Using only one indicator might lead to an underestimate of the influence of adult SEP. Our findings also indicate that multiple pathways might be involved in how social class gets under the skin. Simultaneously, it indicates that intervention measures aimed at tackling the socioeconomic inequalities in T2DM should consider addressing multiple SEP components and implicated pathways, rather than addressing one pathway only [11,32,33].

Of the four SEP indicators examined, income was most strongly associated with prediabetes, although this association was no longer statistically significant after adjustment for all other SEP indicators. We are not sure about this finding, but it might be related to difficulties regarding the measurement of income. Although broad categories of income were used in the questionnaire, people might not precisely know their household income, or they might report upward when their income is low, or downward, when their income is high, or they find the question impertinent [34,35]. Supporting an ordinal or dose-response model underlying the outcome (prediabetes and T2DM), the reported odds ratios of prediabetes for all SEP indicators were lower than those of T2DM, but still elevated for the lowest SEP group. As low income was still significant in model 3 for prediabetes, the correlations between the SEP indicators themselves might just have been too strong for a remaining significance in model 4 where all SEP indicators were included. This might also have led to underestimated coefficients for income in the models for T2DM.

What do the direct effects of the different SEP components tell us about the pathways and the possible points of action for interventions? Varying levels of SEP might contribute to the different vulnerabilities of diabetes through different pathways. Education could, for example, affect the clustering of diabetes through health literacy [3,36]. The knowledge and skills attained through education may help people to reach a higher level of health consciousness, which, in turn, influence a person’s choice of healthy food and health behaviors [37,38,39]. Occupation foremost reflects an individual’s social class and the prestige of a particular occupation [40]. The occupation–diabetes association could be explained by specific toxic exposures, such as high physical demands, low control, and job strain [41,42,43]. People with a higher income may be more easily able to maintain a healthy lifestyle by purchasing health care products and better nutrition [44,45]. Individuals with low incomes mostly live in poorer areas that may not have much access to leisure-time physical activities, and that has an independent effect on T2DM [46,47]. As material resources can still substantially vary within groups with similar incomes, material deprivation may independently influence the risk of developing T2DM, e.g., indirectly by imposing financial constraints on healthy behaviors and, more directly, by affecting living conditions and other factors associated with financial and material disadvantage [48,49]. For example, the lack of a refrigerator or freezer at home may result in people consuming less fruit and vegetables.

Our study supports the hypothesis that SEP in its relation to T2DM and its precursor prediabetes is a multifactorial risk indicator. As stated before, the use of one indicator might underestimate the full influence of adult SEP [50,51]. Our findings indicate the importance of combining the multiple components of what social stratification is about. The different components clearly have the potential of exerting independent, unique effects on health [52,53,54,55]. When we computed a compositional SEP indicator by averaging the four SEP indicators, participants with the lowest compositional SEP had a strikingly 3.37 higher odds of T2DM compared to their better-off counterparts. Hence, our findings indicate that when SEP is the primary determinant, one should consider either studying more than one indicator or use a composite measure of SEP. Furthermore, in order to avoid residual confounding by SEP, researchers should be hesitant to control for education only, as it is unlikely to fully capture the potential confounding by SEP.

Regarding the key toxic component of SEP and its proximate pathways towards T2DM, we were primarily interested in direct effects, but had to acknowledge that the four adulthood socioeconomic indicators were also mutually related along the life course (education impacting occupation impacting income impacting material circumstances). Education remaining such a strong correlate of T2DM in our analyses is, therefore, striking. Our approach foremost allows the identification of the remaining direct effects and gives clues on the etiological pathways and related opportunities for intervention.

Previous studies have confirmed the view that the important foundations of adult health can be found in early childhood [56]. In a previous study, we have confirmed the independent influence of early life socioeconomic conditions as well [17]. In the current study, adding childhood SEP into the analyses did not dramatically attenuate the association between adulthood SEP and T2DM. Accordingly, our findings indicate the independent importance of adulthood SEP for diabetes. In addition, others have argued that education may simply serve as a marker for intelligence in health disparities. However, the inclusion of verbal intelligence in the analyses did not strongly weaken the association between education and T2DM. Although one would perhaps need other (non-verbal) measures of intellectual abilities (and measured in earlier life), our results might challenge the view on intelligence as the fundamental cause of T2DM inequalities [57].

There were some methodological issues. First, the study had a cross-sectional design, which limits our understanding of causality underlying the association between the SEP indicators and prediabetes and T2DM. However, educational level is less vulnerable to reversed causation as education is typically completed in late adolescence or early adulthood [58]. Birth cohorts and advanced longitudinal methods are clearly needed. Second, correlations among the four SEP indicators are likely to cause multicollinearity. However, correlations between the SEP indicators were 0.58 at the highest and in the analyses no convergence problems were encountered. Third, participants who were excluded from the analyses (42%) because of missing data were significantly older (M = 61.21, SD = 8.21 vs. M = 58.81, SD = 8.14), and were more likely to have T2DM (35.8 vs. 23.2%) and to belong to the lower group in terms of educational level, occupational level, and material deprivation (42.0 vs. 26.5%, 41.4 vs. 27.6%, and 37.9 vs. 31.2%, respectively). This pattern may have resulted in underestimated associations. As the measure of occupation excluded people who never had a paid profession (who more often were women), we, strikingly, did not find more women among the 42% with missing data. Fourth, health behaviors, such as dietary patterns and physical activity, were not controlled for, since we perceived these factors as shaped by the socioeconomic context. Controlling for health behaviors would thus have resulted in over-adjustment of the SEP—T2DM associations [44]. Finally, although we examined multiple, individual-level SEP indicators, we have neglected any influences of the socioeconomic characteristics of the neighborhoods in which participants lived [59]. The upcoming work will thus study the contextual influence of the residential environment.

## 5. Conclusions

Our findings suggest that low educational level, occupational level, and material deprivation have independent influences on T2DM. A lower income was associated with prediabetes, although this association was no longer significant when in the analysis all SEP indicators were corrected for each other. In its relation to T2DM and its precursor prediabetes, these data confirm SEP as a multifaceted concept and indicate the need to measure SEP accordingly. In order to tackle the social gradient in prediabetes and T2DM, one should, therefore, address multiple SEP indicators and their possible pathways.

## Figures and Tables

**Figure 1 ijerph-16-01435-f001:**
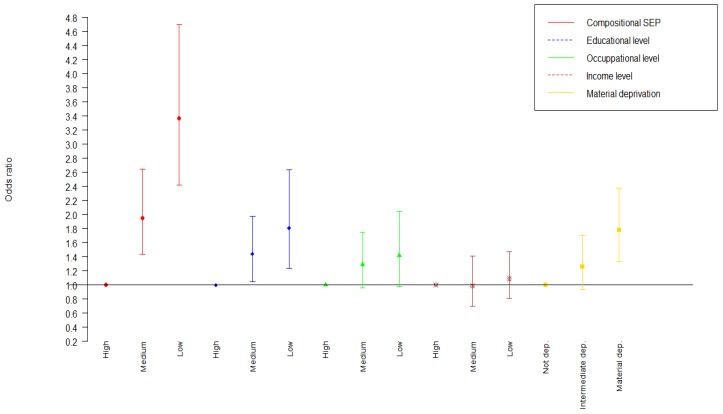
Odds ratios of compositional SEP and separate SEP measures in relation to type 2 diabetes mellitus (T2DM), adjusted for sex, age, early life SEP and verbal intelligence.

**Table 1 ijerph-16-01435-t001:** Demographics, early life socioeconomic position (SEP), educational level, occupational level, income level and material deprivation by diabetes status.

Variables	Diabetes Status			
	Normal Glucose Metabolism (*n* = 1232)	Prediabetes (*n* = 312)	T2DM (*n* = 467)	*p*-Value
Age (mean ± SD)	57.2 ± 8.1	60.9 ± 7.6	61.7 ± 7.6	*p* < 0.001
Male (% ^1^)	44.3	53.5	73.4	*p* < 0.001
Early life SEP (% ^1^)				*p* = 0.001
Low	25.9	31.4	35.3	
Intermediate	30.9	32.1	30.4	
High	43.2	36.5	34.3	
Adulthood SEP				
Educational level (% ^1^)				*p* < 0.001
Low	21.8	29.5	36.6	
Intermediate	29.1	28.2	30.0	
High	49.0	42.3	33.4	
Occupational level (% ^1^)				*p* < 0.001
Low	24.7	27.9	35.3	
Intermediate	35.7	36.2	35.8	
High	39.6	35.9	28.9	
Income level (% ^1^)				*p* = 0.002
Low	26.8	33.3	33.3	
Intermediate	37.3	33.3	40.0	
High	36.0	33.3	27.0	
Material deprivation (% ^1^)				*p* < 0.001
Deprived	27.8	31.7	39.8	
Intermediate deprived	23.1	21.5	21.4	
Not deprived	49.2	46.8	38.8	

^1^ Column percentages.

**Table 2 ijerph-16-01435-t002:** Odds ratios of prediabetes by educational level, occupational level, income level and material deprivation.

Adulthood SEP	Category	Model 1 ^1^				Model 2 ^2^				Model 3 ^3^				Model 4 ^4^			
		Categorical OR	*p*	Standardized OR	*p*	Categorical OR	*p*	Standardized OR	*p*	Categorical OR	*p*	Standardized OR	*p*	Categorical OR	*p*	Standardized OR	*p*
Educational Level	High	Ref															
	Intermediate	1.25	0.15			1.24	0.17			1.24	0.19			1.16	0.41		
	Low	1.41	0.03	0.87	0.03	1.39	0.05	0.90	0.11	1.38	0.08	0.90	0.18	1.22	0.36	0.95	0.63
Occupational level	High	Ref															
	Intermediate	1.26	0.13			1.25	0.14			1.24	0.17			1.14	0.44		
	Low	1.34	0.08	0.88	0.06	1.31	0.11	0.91	0.15	1.28	0.18	0.91	0.22	1.06	0.79	0.97	0.76
Income level	High	Ref															
	Intermediate	0.97	0.86			0.96	0.80			0.95	0.77			0.90	0.54		
	Low	1.49	0.01	0.84	0.01	1.45	0.02	0.86	0.03	1.42	0.04	0.86	0.03	1.28	0.20	0.89	0.14
Material deprivation	Not deprived	Ref															
	Intermediate deprived	1.08	0.64			1.08	0.64			1.07	0.67			1.03	0.86		
	Material deprived	1.30	0.08	0.89	0.08	1.26	0.12	0.91	0.14	1.25	0.15	0.91	0.16	1.11	0.52	0.96	0.56

^1^ Adjusted for sex and age; ^2^ Adjusted for sex, age, and early life SEP; ^3^ Adjusted for sex, age, early life SEP, and verbal intelligence; ^4^ Adjusted for sex, age, early life SEP, verbal intelligence and all four SEP indicators.

**Table 3 ijerph-16-01435-t003:** Odds ratios of T2DM by educational level, occupational level, income level and material deprivation.

Adulthood SEP	Category	Model 1 ^1^				Model 2 ^2^				Model 3 ^3^				Model 4 ^4^			
		Categorical OR	*p*	Standardized OR	*p*	Categorical OR	*p*	Standardized OR	*p*	Categorical OR	*p*	Standardized OR	*p*	Categorical OR	*p*	Standardized OR	*p*
Educational Level	High	Ref															
	Intermediate	1.73	<0.001			1.79	<0.001			1.68	<0.001			1.44	0.02		
	Low	2.53	<0.001	0.66	<0.001	2.64	<0.001	0.68	<0.001	2.32	<0.001	0.69	<0.001	1.81	<0.01	0.80	0.01
Occupational level	High	Ref															
	Intermediate	1.65	<0.001			1.65	<0.001			1.56	<0.01			1.29	0.09		
	Low	2.33	<0.001	0.67	<0.001	2.38	<0.001	0.69	<0.001	2.03	<0.001	0.72	<0.001	1.42	0.07	0.84	0.03
Income level	High	Ref															
	Intermediate	1.52	<0.001			1.51	<0.001			1.40	0.02			0.99	0.97		
	Low	1.97	<0.001	0.72	<0.001	1.91	<0.001	0.74	<0.001	1.65	<0.01	0.78	<0.001	1.09	0.56	0.96	0.55
Material deprivation	Not deprived	Ref															
	Intermediate deprived	1.37	0.04			1.37	0.04			1.33	0.07			1.26	0.14		
	Material deprived	2.19	<0.001	0.71	<0.001	2.10	<0.001	0.73	<0.001	1.96	<0.001	0.75	<0.001	1.78	<0.001	0.80	<0.001

^1^ Adjusted for sex and age; ^2^ Adjusted for sex, age, and early life SEP; ^3^ Adjusted for sex, age, early life SEP, and verbal intelligence; ^4^ Adjusted for sex, age, early life SEP, verbal intelligence and all four SEP indicators.

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
