# Peer review of "Adulthood Socioeconomic Position and Type 2 Diabetes Mellitus—A Comparison of Education, Occupation, Income, and Material Deprivation: The Maastricht Study"

_ijerph, 2019, doi:10.3390/ijerph16081435_

Round 1

Reviewer 1 Report

The manuscript presents a study examining the relationship between several indicators of socio-economic status (SES) and prediabetes and type 2 diabetes (T2DM) in a sample of middle-to-older age adults from the Netherlands. The sample’s gender ratio is more or less balanced. However, there are considerably more men than women among persons with clinically diagnosed with T2DM. Although there are more participants with normal glucose metabolism than those with T2DM and prediabetes, the descriptive statistics show an approximately normal distribution of participants along the lines of socio-economic indicators. Thus, there are almost equal proportions of participants who were (dis)advantaged in terms of educational status, occupational status and income. I appreciated that the analyses were grounded in relevant theories and were also informed by existing empirical data. I believe the authors went as far as they could to discuss/explain their findings without going beyond what their data allowed. 

However, this topic is not novel, and the results are hardly unexpected. Thus, the authors should emphasize how their study is different from the abundant empirical literature on the relationship between SES and diabetes morbidity and how this work advances our understanding of social risk factors for diabetes.

Author Response

Author's Reply to the Review Report-IJERPH-479015

Reviewer #1, Point #1: However, there are considerably more men than women among persons with clinically diagnosed with T2DM. Although there are more participants with normal glucose metabolism than those with T2DM and prediabetes, the descriptive statistics show an approximately normal distribution of participants along the lines of socio-economic indicators. Thus, there are almost equal proportions of participants who were (dis)advantaged in terms of educational status, occupational status and income.

We thank the Reviewer for this detailed comment. It is not fully clear what was the reviewer referring to. Our sample was stratified according to diabetes status as reported on p.2, line 66-71. Therefore, we used column percentages in Table 1 (p.4), as row percentages would suggest cumulative incidence rates which they clearly are not. Column percentages avoid this problem. To clarify this issue, we added a footnote to Table 1 on p.5, line 164 indicating that the percentages are column percentages.  

Reviewer #1, Point #2: However, this topic is not novel, and the results are hardly unexpected. Thus, the authors should emphasize how their study is different from the abundant empirical literature on the relationship between SES and diabetes morbidity and how this work advances our understanding of social risk factors for diabetes.

We thank the Reviewer for this comment. To better address our novelty, we added a statement in the discussion section as follows (p.9, line 202-203).

"To our knowledge, this is the first study to compare, in a single sample, the relative strengths of the association of various SEP indicators with objectively measured T2DM.”

Reviewer 2 Report

Very well written. Only a few minor edits. 

Author Response

Author's Reply to the Review Report-IJERPH-479015

Reviewer #2, Point #1: Very well written. Only a few minor edits.

We thank the Reviewer for this comment.

Reviewer #2, Point #2: Typo page 1, line 30: therefor  therefore

We thank the Reviewer for noticing this.

We changed the sentence to the following on p. 1, line 30: “In order to tackle the social gradient in prediabetes and T2DM, 29 one should therefore address multiple SEP indicators and their possible pathways.”

Reviewer #2, Point #3: The use of abbreviation, page 2, line 65

We thank the Reviewer for noticing this.

We changed the sentence to the following on p. 2, line 65: “Recruitment was stratified according to known T2DM status for reasons of efficiency.”

Reviewer #2, Point #3: Write the words then use the abbreviation thereafter. This applies to the remaining abbreviations.

We thank the Reviewer for this comment.

We changed the sentence to the following on p. 2, line 74-76: “T2DM was defined according to the World Health Organization (WHO) diagnostic criteria of glucose tolerance status [25]. All participants underwent a standardised 7-point oral glucose tolerance test (OGTT) after overnight fasting.”
